# Therapeutic Options and Outcomes for the Treatment of Children with Gram-Positive Bacteria with Resistances of Concern: A Systematic Review

**DOI:** 10.3390/antibiotics12020261

**Published:** 2023-01-28

**Authors:** Lorenzo Chiusaroli, Cecilia Liberati, Luigi Rulli, Elisa Barbieri, Marica De Pieri, Costanza Di Chiara, Daniele Mengato, Carlo Giaquinto, Daniele Donà

**Affiliations:** 1Pediatric Infectious Diseases, Department for Women’s and Child’s Health, University Hospital of Padova, 35128 Padua, Italy; 2Hospital Pharmacy Department, University Hospital of Padova, 35128 Padua, Italy

**Keywords:** children, neonates, Gram-positive bacteria, multidrug resistant, methicillin-resistant *Staphylococcus aureus*, vancomycin-resistant *Enterococci*, outcome assessment, treatment, review

## Abstract

Background: Methicillin-resistant *Staphylococcus aureus* (MRSA), methicillin-resistant coagulase-negative *Staphylococci* (MR-CoNS), and vancomycin-resistant *Enterococci* (VRE) are increasing worldwide and represent a threat for the limited treatment options in pediatric patients and neonates compared to adults. Recommendations in pediatrics are mainly extrapolated from adults’ studies. Methods: A literature search for the treatment of these pathogens in children (<18 years old) was conducted in Embase, MEDLINE, and Cochrane Library. Studies reporting data on single-patient-level outcomes related to a specific antibiotic treatment for multidrug resistant (MDR) Gram-positive bacterial infection in children were included. Studies reporting data from adults and children were included if single-pediatric-level information could be identified (PROSPERO registration: CRD42022383867). Results: The search identified 11,740 studies (since January 2000), of which 48 fulfilled both the inclusion and the exclusion criteria and were included in the analysis: 29 for MRSA, 20 for VRE, and seven for MR-CoNS. Most studies were retrospective studies. Vancomycin was mainly used as a comparator, while linezolid and daptomycin were the most studied antimicrobials showing good efficacy. Conclusions: Linezolid showed a safety and efficacy profile in a neonatal setting; daptomycin is increasingly used for MRSA, but the evidence is scarce for VRE.

## 1. Introduction

Gram-positive multidrug-resistant organisms (GP-MDROs) significantly cause hospital-related infections in neonatal and pediatric populations. The increased life expectancy for chronically ill individuals is accompanied by the increased use of invasive devices and access to medical services, raising the risk of colonization and infection with MDR organisms. 

Due to their clinical and public health impact, the World Health Organization (WHO) and Centers for Disease Control and Prevention (CDC) have listed both methicillin-resistant *Staphylococcus aureus* (MRSA) and vancomycin-resistant *Enterococcus* (VRE) as high-priority pathogens in urgent need of drug research and development [1,2]. In Europe, in 2019, 18.6% of *Staphylococcus aureus* (SA) isolates showed single or combined resistances with significant geographical variability; methicillin resistance (MR) can be isolated (2.7%) or combined with quinolones (9.6%) or quinolones and rifampicin (0.6%) [3]. The burden of MRSA, overall, increased from 2007 and was higher in infants younger than 12 months, compared to other age groups [3,4].

The WHO surveillance net in low–middle-income countries (LMICs) reports relatively higher rates of MR for SA compared to high-income countries (HICs): 33.3% in LMICs and 15% in HICs [5].

Reports from HICs (USA) showed that MRSA infections occur mainly as healthcare-associated infections in predisposed individuals, but community-acquired (CA) infections are increasing [6]. In African countries, it is estimated that the majority of MRSA infections are hospital-acquired, but significantly under-recorded cases make the CA-MRSA reservoir underrated [7,8].

In Europe, infections and deaths attributable to VRE doubled between 2007 and 2015 [3,9]. Proportions of bloodstream infections caused by VR *Enterococcus faecium* (VREFm) increased from 8.1% in 2012 to 19.0% in 2018. The ECDC reported a significant increase in the percentage of VRE isolated in Europe, from 11.6% in 2016 to 16.8% in 2020, in the overall population [3]. However, children and adolescents showed lower VRE proportions than older age groups [10].

Compared to other MDR organisms, a relationship between increased resistance rates for *E. faecium* and country income status is not observed [11], although the presence of VRE has been widely described in Africa and South America [12,13].

Coagulase-negative staphylococci (CoNS) are typically resistant to methicillin and multiple drugs due to the selective pressure of antibiotic exposure, with an increasing trend [14,15,16]. Although CoNS harbor fewer virulence factors and do not correlate with high morbidity and mortality, they are a significant cause of sepsis in neonatal intensive care units, representing a challenge for the limited antibiotic options in this population [17,18].

To date, the management of GP-MDROs is based on a few indications extrapolated from data on adults, with scant evidence in pediatrics.

This study aims to critically appraise the current antimicrobial treatment options and the relative outcomes for infections caused by the most common Gram-positive bacteria harboring resistances of concern for treatment in the pediatric and neonatal populations: MRSA, VRE, and MR-CoNS.

## 2. Methods

### 2.1. Literature Search

This systematic review was carried out according to the Preferred Reporting Items for Systematic Reviews and Meta-Analyses (PRISMA) guidelines (Figure 1). Embase, Medical Literature Analysis and Retrieval System Online (MEDLINE), and Cochrane Library were searched for relevant studies, combining Medical Subject Heading (MeSH) and free-text terms for “children”, “Gram-positive bacteria”, “resistant”, “methicillin-resistant *Staphylococcus aureus*”, “vancomycin-resistant *Enterococci*”, and “outcome assessment” (see complete search strategy in Appendix A. The search strategy involved restrictions on the date (from 1 January 2000 to 1 November 2022) but not on language. All studies on children younger than 18 were considered.

This study was registered with the International Prospective Register of Systematic Reviews (PROSPERO) under record number CRD42022383867.

We included studies with any method of diagnosing infections with pathogens of interest in children, neonates and preterms. All sites of infection were included. The search results were exported to Rayyan software for further manuscript assessment and handling.

### 2.2. Study Selection

Assessments of the titles, abstracts, and full texts were conducted independently by three investigators (L.C., C.L., and L.R.). Discussion with a fourth senior reviewer (D.D.) resolved any disagreement regarding study selection.

### 2.3. Eligibility Criteria

Eligible study designs included randomized clinical trials, observational studies, prospective or retrospective designs, concomitant or historical control studies, case series, and case reports. Meta-analyses, systematic reviews, and narrative reviews were not included. Studies investigating any antimicrobial treatment for infections caused by the following bacteria were included: MRSA, VRE, and MR-CoNS.

The populations of interest were children, as well as term and preterm newborns, with confirmed GP-MDROs infections receiving antimicrobial treatment and presenting clinical and/or microbiological outcomes. 

The outcomes of interest we collected from the selected studies were infection-related mortality from the initiation of treatment until discharge, clinical success (defined as complete resolution or a substantial improvement in signs and symptoms of the index infection), and microbiological success (measured by the suppression, eradication, or relapses of bacterial growth). 

Studies published between 1 January 2000 and 30 October 2022 were included. Further details are reported in the PICOS format (P: problem/patient/population; I: intervention; C: comparison/control; O: outcome; S: study design). 

P (Participants/population): Children, as well as term and preterm newborns, with confirmed GP-MDROs infections that were receiving antimicrobial treatment and presenting clinical and/or microbiological outcomes clearly specified.

I (Intervention/Exposure): Any antimicrobial treatment clearly defined.

C: (Comparator/Control): Standard of care at the time and place where the study was conducted. Not applicable.

O: (outcomes): The primary outcome was mortality-related GP-MDRO infection. Secondary outcomes were clinical success, defined as complete resolution or substantial improvement of signs and symptoms of the index infection, microbiological success measured by suppression, eradication, or relapses of bacterial growth, and treatment-related adverse effects.

### 2.4. Data Extraction and Assessment of Study Quality

The following data were extracted using a standardized data collection form: Study characteristics (authors, year of publication, study design, study location, and country);Patient characteristics (age, care setting, and inclusion and exclusion criteria);Type of MDR;Setting;Main results with accuracy measures;Health outcomes (e.g., mortality, clinical response, and microbiological eradication);Main results.


Standardized predetermined study criteria were applied to all full-text documents. The selection process is presented in Figure 1.

The quality and risk of bias in individual studies were jointly assessed at the study and outcome level by all reviewing authors using the Study Quality Assessment Tool from The National Heart, Lung, and Blood Institute [19]. The quality assessment results are presented in Table 1, Table 2 and Table 3. Each study was quality-rated according to one of the following categories on the basis of the proportion of yes answers to all relevant questions: poor quality, 0–40%; fair quality, 41–80%; good quality, 81–100% [19].

### 2.5. Summary Measures

The following measures of treatment success were included: absolute values, absolute risk differences, hazard ratio (HR), relative risk, and odds ratio. Unadjusted and adjusted measures were included if available.

## 3. Results

### 3.1. Study Selection

A total of 11,740 records were found, and 2225 duplicate records were removed. Then, 9515 records were screened and excluded by title or abstract, followed by 138 exclusions after reading the full text as they did not meet the eligibility criteria. Forty-two articles regarding Gram-negative bacteria were also excluded. Therefore, 48 articles were included in the systematic qualitative review (Figure 1). We report the characteristics of the studies in Table 1, Table 2 and Table 3. Figure 2 shows the studies with a relevant number of patients.

### 3.2. Methicillin-Resistant Staphylococcus aureus (MRSA)

We included 29 studies regarding MRSA: 11 randomized controlled trials, two prospective studies, six retrospective studies, two case series, and nine case reports [20,21,22,23,24,25,26,27,28,29,30,31,32,33,34,35,36,37,38,39,40,41,42,43,44,45,46,47,48].

Clinical and microbiological success is reported in Table 1.

Most relevant studies were conducted in high-income countries (19/30), and three were multicenter international studies. The selected studies reported the following MRSA clinical infections: skin and skin structure infections (SSSIs), bloodstream infections (BSIs), central line-associated bloodstream infections (CLABSIs), endocarditis, community and hospital pneumonia, bone infections, and others.

The included antibiotic regimens were linezolid (13/30), daptomycin (10/30), ceftaroline (3/30), and other drugs 4/30 (TMP-SMX, cephalexin, and quinupristin/dalfopristin). Vancomycin was considered the comparator in 4/30 studies.

### 3.3. Vancomycin-Resistant Enterococcus (VRE)

We included 20 studies regarding VRE: one randomized controlled trial, two prospective studies, three retrospective studies, and nine case reports [22,23,25,27,36,46,49,50,51,52,53,54,55,56,57,58,59,60,61,62].

Clinical and microbiological success and antimicrobial safety are reported in Table 2.

The settings were different, but most studies (11/20) were conducted in HICs. The selected studies reported the following infections: BSIs, central nervous system (CNS), pneumonia, and other sites.

The included antibiotic regimens were linezolid (11/20), daptomycin (6/20), and other drugs 5/20 (tigecycline, quinupristin/dalfopristin). Linezolid as monotherapy was the more described antibiotic. A case report described the use of phage therapy for VRE [54].

### 3.4. Methicillin-Resistant Coagulase-Negative Staphylococci (MR-CoNS)

We included seven studies regarding MR-CoNS: one randomized controlled trial, one retrospective study, one case series, and three case reports [25,27,46,63,64,65,66].

Clinical and microbiological success and antimicrobial safety are reported in Table 3.

Only two out of six studies were conducted in LMICs. The selected studies reported the following infections: BSIs, CNS, pneumonia, and other sites.

Described antimicrobials were linezolid, vancomycin, and daptomycin.

**Table 1 antibiotics-12-00261-t001:** Treatment choices and outcomes for methicillin-resistant *Staphylococcus aureus* (MRSA).

Reference	Study Type	Publication Year	Country	Center	Setting	N of Patients (Inc/All)	Median Age (Year)	Resistance	Site of Infection	Antimicrobial Treatment	Route	Outcomes Measures	Outcomes Measures	Results	Quality Assessment
**Korczowski, Bartosz** [20]	Randomized, observer-blinded, active-controlled	2016	Worldwide	Multicenter	Inpatient	25	Not indicated ^#^	MRSA	SSSIs	Ceftaroline	iv	Clinical success, microbiological eradication	Absolute value	**Clinical success:** ceftaroline 17/18 (94%) vs. vancomycin 6/7 (86%) **Microbiological response:** 16/18 (89%) ceftaroline vs. 4/7 (57%) vancomycin	Good
John S. Bradley [21]	Randomized, controlled, double-blind	2020	Worldwide	Multicenter	Inpatient, outpatient	8	9	MRSA	Bone	Daptomycin	iv	Clinical success, microbiological eradication	Absolute value	**Clinical success:** daptomycin 2/4 (50.0%) vs. vancomycin1/4 (25.0%) **Microbiological eradication:** daptomycin 2/4 (50.0%) vs. vancomycin 3/4 (75.05)	Good
Monica I. Ardura [22]	Retrospective	2007	USA	Monocenter	Inpatient, outpatient	14	6	MRSA	BSIs, endocarditis, pneumonia, pyomyositis, and osteoarthritis	Daptomycin	iv	Clinical success	Absolute value	**Clinical success:** daptomycin 13/14 (92%)	Fair
**Maria Moschovi** [23]	Prospective	2010	Greece	Monocenter	Oncoematological	4	4	MRSA	CLABSIs	Linezolid	iv	Clinical success	Absolute value	**Clinical success:** 4/4 (100%)	Fair
Sheldon L. Kaplan [25]	Randomized trial, comparator-controlled	2003	USA e Mexico	Multicenter	Inpatient, outpatient	29	6	MRSA	HAP, SSSIs, BSIs	Linezolid vs. vancomycin	iv	Clinical success	Absolute value	**Clinical success** Linezolid 16/17 (94%) vs. vancomycin 9/12 (90%)	Good
Glenn Isaacson [26]	Retrospective	2008	USA	Monocenter	Outpatient	7	2	MRSA	Ear	Linezolid	Oral	Clinical success	Absolute value	**Clinical success:** 100% (7/7)	Poor
Adem YılmAz [27]	Retrospective	2010	Turkey	Monocenter	Inpatient	1	11	MRSA	CNS	Linezolid	iv	Clinical success	Absolute value	**Clinical success:** 1/1	Fair
Tae-Jung Sung [28]	Case report	2008	Korea	Monocenter	Inpatient	1	Premature	MRSA	Endocarditis	Linezolid	iv	Clinical success	Absolute value	**Clinical response:** 0/1	Poor
Joshua I. Chan [29]	Case report	2020	USA	Monocenter	Inpatient	1	Premature	MRSA	BSIs	Daptomycin	iv	Clinical success	Absolute value	**Clinical success:** 1/1	Poor
Zakaria Jalal [30]	Case report	2013	Turkey	Monocenter	Inpatient	1	12	MRSA	Endocarditis, BSIs	Daptomycin	iv	Clinical success	Absolute value	**Clinical success:** 1/1	Poor
Aaron e. Chen [31]	Randomized trial	2010	USA	Monocenter	Outpatient	133	Not indicated ^#^	MRSA	SSSIs	Cephalexin Vs Clindamycin	Oral	Clinical success	Absolute value	**Clinical success:** cephalexin 63/63 (100%) vs. clindamycin 66/70 (94%)	Good
Stefan Borgmann [32]	Case report	2016	Germany	Monocenter	Inpatient	1	9	MRSA	BSIs	Ceftaroline	iv	Clinical success	Absolute value	**Clinical success:** BSIs resolved	Poor
Al Zabem [33]	Case series	2016	Jordan	Monocenter	Inpatient, outpatient	5	5.8	MRSA	Bone	Vancomycin + rifampicin, then Tmp/Smx + rifampicin	iv/oral	Clinical success	Absolute value	**Clinical success:** 4/5 (80%)	Poor
Lucy Holmes [34]	Randomized trial	2015	USA	Monocenter	Inpatient	137	Not indicated ^#^	MRSA	SSSIs	TMP-SMX	Oral	Clinical success	Absolute value	**Clinical success:** 61/68 (89%) 3 days of therapy vs. 68/69 (98%) 10 days of therapy	Good
Satoshi Iwata [39]	Open-label, single-arm phase 2 study	2021	Japan	Multicenter	Inpatient, outpatient	8	7	MRSA	cSSIs, BSIs	Daptomycin	iv	Clinical success	Absolute value	**Clinical success:** cSSIs 5/7 (71%), BSI 1/1 (100%)	Good
Nicholas M. Fusco [37]	Retrospective	2019	USA	Monocenter	Cystic fibrosis	122 *	18	MRSA	Pneumonia	Linezolid vs. vancomycin	iv	Clinical success, adverse effects	Absolute value	**Clinical success:** vancomycin 53/66 (80.3%) vs. linezolid 50/66 (76%) **Adverse effect:** vancomycin 10/66 (15.2%) vs. linezolid 2/66 (3%)	Good
John Bradley [38]	Randomized trial	2017	Worldwide	Multicenter	Inpatient, outpatient	97	Not indicated ^#^	MRSA	SSSIs	Daptomycin vs. standard of care	iv	Clinical success, adverse effects	Absolute value	**Clinical success:** daptomycin 82/97 (85%) and SOC 41/46 (89%) **Adverse effect:** 14% of daptomycin vs. 17% of SOC	Good
Aaron Cook [40]	Case report	2005	USA	Monocenter	Inpatient	1	4	MRSA	CNS	Linezolid + rifampicin	iv	Clinical and microbiological success	Absolute value	**Clinical success and microbiological success:** 1/1	Poor
masayoshi shinjoh [35]	Retrospective	2012	Japan	Monocenter	Inpatient	16	4	MRSA	BSIs, skin, lung, CNS	Linezolid	iv/oral	Clinical success	Absolute value	**Clinical success:** 10/16 (63%)	Fair
Loeffler A. [36]	Prospective	2002	USA	Multicenter	Not specified	8	7	MRSA	BSIs, skin, pneumonia, joints, bone, CLABSI	Quinupristin/dalfopristin	iv	Clinical success	Absolute value	**Clinical success:** 5/8 (62%)	Fair
**Antonio c. Arrieta** [48]	Randomized multicenter	2018	Worldwide	Multicenter	Not specified	10	8	MRSA	Bone, joints, BSIs, CLABSIs, intrabdominal	Daptomycin vs. SOC	iv/oral	Clinical success	Absolute value	**Clinical success:** daptomycin 6/7 (85%) vs. 2/3 (67%) SOC	Good
Kenneth Wible [24]	Randomized controlled trial	2003	USA	Multicenter	Not specified	20	10	MRSA	SSSIs	Linezolid	oral	Clinical success	Absolute value	**Clinical success:** linezolid 12/13 (92%) vs. cefadroxil 6/7 (85%)	Good
Ram Yogev [41]	Open label, randomized	2003	USA	Multicenter	Inpatient	18	3	MRSA	SSSIs	Linezolid vs. vancomycin	iv/oral	Clinical success	Absolute value	**Clinical success:** linezolid 9/10 (90%) vs. vancomycin 6/8 (75%)	Good
Gallagher [42]	Case report	2008	UK	Monocenter	Inpatient	1	4	MRSA-VISA	CNS	Rifampicin, linezolid	iv/oral	Clinical and microbiological success	Absolute value	**Clinical success:** 1/1	Poor
Chih-Jung Chen [43]	Retrospective	2007	Taiwan	Monocenter	Not specified	11	6	MRSA	Bone	Linezolid iv/os	iv/oral	Clinical and microbiological success	Absolute value	**Clinical success:** 9/11 (81%)	Fair
Lara Jacobson [44]	Case report	2009	USA	Monocenter	Inpatient	1	15	MRSA	BSIs	Daptomycin	iv	Clinical success	Absolute value	**Clinical success:** 0/1	Poor
**Salerno** [45]	Case report	2017	USA	Monocenter	Inpatient	1	Premature	MRSA	Pneumonia	Ceftaroline + rifampicin	iv	Clinical and microbiological success	Absolute value	**Clinical and microbiological success:** 1/1	Poor
Hussain [47]	Case report	2011	United Kingdom	Monocenter	NICU	1	Premature	MRSA	BSIs	Daptomycin	iv	Clinical success	Absolute value	**Clinical success:** 1/1	Poor
**Palma** [46]	Case series	2013	Italy	Monocenter	PICU	3	Not indicated	MRSA	BSIs, SSSIs	Daptomycin	iv	Clinical success	Absolute value	**Clinical success:** 3/3	Poor

USA: United States of America; BSI: bloodstream infection; SSSI: skin and skin structure infection; cSSI: complicated skin and skin structure infection; SOC: standard of care; CNS: central nervous system, iv: intravenous, VISA: vancomycin intermediate-resistant *Staphylococcus aureus*, CLABSI: central line-associated bloodstream infection, HAP: hospital-acquired pneumonia, TMP-SMX: trimethoprim–sulfamethoxazole, NICU: neonatal intensive care unit. * Study reported 122 episodes of acute pulmonary exacerbations in 49 patients with cystic fibrosis. We considered the total of episodes because etiology, treatment, and outcome were defined in each one. ^#^ Study regarding pediatric population. The mean age was not clearly defined.

**Table 2 antibiotics-12-00261-t002:** Choice of treatment and outcomes for vancomycin-resistant *Enterococcus* (VRE).

Reference	Study Type	Publication Year	Country	Center	Setting	N of Patients (Inc/All)	Median Age (Year)	Resistance	Bacteria	Site of Infection	Antimicrobial Treatment	Route	Outcomes Measures	OutcoMes Measures	Results	Quality Assessment
Monica I. Ardura [22]	Retrospective	2007	USA	Monocenter	Inpatient	1	10	VRE	*E. faecium*	UTIs	Daptomycin	iv	Clinical success	Absolute value	**Clinical success:** daptomycin 0/1	Fair
**Maria Moschovi** [23]	Prospective	2010	Greece	Monocenter	Oncoematological	10	2.8	VRE	*Enterococcus* spp.	BSIs, stool	Linezolid	iv	Clinical success	Absolute value	**Clinical success:** 10/10 (100%)	Fair
Ayse Şahin [49]	Case report	2019	Turkey	Monocenter	Inpatient	1	2 months	VRE	*E. faecium*	CNS	Linezolid iv + daptomycin iv	iv + ivt	Clinical success	Absolute value	**Clinical success:** 1/1	Poor
Ayse Sahina [50]	Case report	2019	Turkey	Monocenter	Inpatient	1	5 months	VRE	*E. faecium*	CNS	Tigecycline	iv + ivt	Microbiological eradication	Absolute value	**Microbiological eradication:** 1/1	Poor
Heather B. Jaspan [51]	Case report	2010	USA	Monocenter	Inpatient	1	21 months	VRE	*Enterococcus faecium*	CNS	Linezolid + daptomycin + tigecycline + daptomycin IVT	iv + ivt	Clinical success	Absolute value	**Clinical success:** 1/1	Poor
Rene Hoehn [52]	Case report	2006	Germany	Monocenter	NICU	2	Preterm	VRE	*Enterococcus* spp.	BSIs	Linezolid	iv	Clinical success	Absolute value	**Clinical and success** 2/2	Poor
Adem YılmAz [27]	Retrospective	2010	Turkey	Monocenter	Inpatient	1	11	VRE	*E. faecium*	CNS	Linezolid	iv	Clinical success	Absolute value	**Clinical success:** 1/1	Fair
Sheldon L. Kaplan [25]	Trial randomized	2003	USA and central America	Multicenter	Inpatient	3	3	VRE	*E. faecium*	BSIs	Linezolid	iv	Clinical success	Absolute value	**Clinical success:** 2/3 (66%)	Good
**Marco Fossati** [53]	Case report	2010	Italy	Inpatient	Oncoematological	1	11	VRE	*E. faecium*	BSIs	Daptomycin	iv	Clinical success	Absolute value	**Clinical success** 0/1	Poor
Loeffler A. [36]	Prospective	2002	USA	Multicenter	Not specified	101	7	VRE	*Enterococcus* spp.	BSIs, skin, pneumonia, joint, bone, CLABSI	Quinupristin/dalfopristin	iv	Clinical success	Absolute value	**Clinical success:** 71/101 (70%)	Fair
Kevin Paul [54]	Case report	2021	Germany	Monocenter	Inpatient	1	10 months	VRE		Abdominal	Bacteriophage therapy	iv	Clinical success	Absolute value	**Clinical success:** 1/1	Poor
James W. Gray [55]	Case Series	2000	UK	Monocenter	Inpatient and outpatient	8	7	VRE	*Enterococcus* spp.	BSIs, abdominal	Quinupristin/dalfopristin	iv	Clinical success	Absolute value	**Clinical success:** 7/8 (87%)	Fair
Jocelyn Ang [56]	Case report	2003	USA	Monocenter	NICU	1	Premature	VRE	*E. faecium*	Endocarditis	Linezolid	iv	Clinical and microbiological success	Absolute value	**Clinical and microbiological success:** 1/1	Poor
Mehmet Baysallar [57]	Case report	2006	Turkey	Monocenter	Inpatient	1	7 months	VRE	*E. faecium*	CNS	Chloramphenicol, rifampin, and meropenem	iv	Clinical and microbiological success	Absolute value	**Clinical and microbiological success:** 1/1	Poor
**M. Travaglianti** [58]	Retrospective	2007	Argentina	Monocenter	Inpatient	15	7 years	VRE	*Enterococcus* spp.	BSIs, UTIs, abdominal, endocarditis	Linezolid	iv/oral	Clinical and microbiological success	Absolute value	**Clinical and microbiological success:** 13/15 (87%)	Poor
Graham [59]	Case report	2002	USA	Monocenter	NICU	1	Preterm	VRE	*E. faecium*	CNS	Linezolid	iv	Clinical success	Absolute value	**Clinical success:** 1/1	Poor
Beneri [60]	Case report	2008	USA	Monocenter	Inpatient	1	Neonate	VRE	*E. faecium*	BSIs	Daptomycin + doxyxyxline	Iv	Clinical success	Absolute value	**Clinical success:** 1/1	Poor
Maranich [61]	Case report	2008	USA	Monocenter	Inpatient	1	17 months	VRE	*E. faecium*	CNS	Linezolid	Iv	Clinical success	Absolute value	**Clinical success:** 1/1	Poor
**Palma** [46]	Case series	2013	Italy	Monocenter	PICU	1	Not indicated	VRE	*E. faecium*	BSIs, SSSIs	Daptomycin	iv	Clinical success	Absolute value	**Clinical success:** 1/1	Poor
Ergaz [62]	Case report	2009	Israel	Monocenter	NICU	3	Preterm	VRE	*E. faecium*	BSIs, CNS	Linezolid	Iv	Clinical success	Absolute value	**Clinical success:** 3/3	Poor

CNS: central nervous system. IV: intravenous; IVT: intraventricular USA: United States of America. VRE: Vancomycin-resistant *enterococci*. UTI: urinary tract infection, CLABSI: central line-associated bloodstream infection, BSI: bloodstream infection, VAP: ventilator-associated pneumonia, NICU: neonatal intensive care unit, SSSI: skin and skin structure infection, UTI: urinary tract infection.

**Table 3 antibiotics-12-00261-t003:** Choice of treatment and outcomes for methicillin-resistant coagulase-negative *staphylococci* (MR-CoNs).

Reference	Study Type	Publication Year	Country	Center	Setting	N of Patients (Inc/All)	Median Age (Year)	Resistance	Bacteria	Site of Infection	Antimicrobial Treatment	Route	Outcomes Measures	Outcomes Measures	Results	Quality Assessment
Shanti [63]	Case report	2009	Malesia	Monocenter	Inpatient	1	1	MR	*S. epidermis*	CNS	iv teicoplanin + IVT teicoplanin 10 mg daily	iv+ivt	Microbiological eradication	Absolute value	**Microbiological eradication:** 1/1	Poor
Sheldon L. Kaplan [25]	Trial randomized	2003	USA e central America	Multicenter	Inpatient	46	2	MR	*S. epidermis*	BSI	Linezolid vs. vancomycin	iv	Clinical success	Absolute value	**Clinical success:** linezolid 29/34 (85.3) vs. vancomycin 10/12 (83.3)	Good
Adem YılmAz [27]	Retrospective	2010	Turkey	Monocenter	Inpatient	4	11	MR	*S. epidermidis*	CNS	Linezolid	iv	Clinical success	Absolute value	**Clinical success:** 4/4	Fair
**C. Minotti** [64]	Case report	2022	Italy	Monocenter	NICU	1	Preterm	MR	*S. epidermidis*	CLABSI	Daptomycin	iv	Clinical success	Clinical success	**Clinical success:** 1/1	Poor
**Fumihiro ochi** [65]	Case series	2018	Japan	Monocenter	Inpatient	2	2	MR	*S. epidermidis*	CNS	Linezolid	iv	Clinical success	Absolute value	**Clinical success:** 2/2	Fair
**Palma** [46]	Case series	2013	Italy	Monocenter	PICU	3	Not indicated	MR	*S. epidermidis*	BSI, SSSIs	Daptomycin	iv	Clinical success	Absolute value	**Clinical success:** 3/3	Poor
Gawronski [66]	Case report	2015	Ohio	Monocenter	NICU	1	Preterm	MR	*S. epidermidis*	BSI	Daptomycin	iv	Clinical success	Absolute value	**Clinical success:** 1/1	Poor

CLABSI: central line-associated bloodstream infection; SOC: standard of care, MR: methicillin-resistant, CNS: central nervous system, iv: intravenous, IVT: intraventricular, CoNS: coagulase-negative staphylococci, CLABSI: central line-associated bloodstream infection, NICU: neonatal intensive care unit.

## 4. Discussion

Gram-positive bacteria may harbor several types of resistance to one or more antimicrobial class agents, with specific criteria to be fulfilled for being defined as MDR [67]. This research focused on the treatment of the most common drug-resistant Gram-positive pathogen infections.

### 4.1. MRSA, VRE

We found most studies (21/27) on MRSA, predominantly analyzing its treatment in invasive infections (SSSIs, bloodstream, bone, pneumonia, and CLABSIs).

Only one retrospective study on uncomplicated SSSIs caused by MRSA was included, in which cephalexin was as effective as clindamycin. However, as cephalexin does not exert any activity against MRSA, the authors concluded that wound care and drainage are more important than the antimicrobial choice [31].

For invasive infections, vancomycin, which is usually considered and recommended as first-line treatment for MRSA infections in pediatrics [68,69], was mainly used in our research as a comparator to assess the efficacy of other molecules: daptomycin, ceftaroline, and linezolid. We found no studies (except a case report) on teicoplanin, as it is not approved by the FDA and is used mainly in European countries.

Ceftaroline is currently approved by FDA and EMA for children and newborns to treat cSSSIs and community-acquired pneumonia [70,71]; however, efficacy data for MR organisms in pediatrics remain scarce. In our research, ceftaroline was evaluated for community-acquired MRSA (CA-MRSA) in children with SSSIs compared with vancomycin in 25 patients, resulting in a similar efficacy rate [20]. However, children with complicated infections or bacteremia were excluded from the study. In a case report, ceftaroline was successfully used (in combination therapy with rifampicin) to treat MRSA bloodstream infection and pneumonia in a preterm infant [45]. Ceftaroline showed a favorable efficacy and safety profile in newborns with late-onset sepsis in a case series, but MRSA was not considered, and antibiograms of CoNs were not provided [72]. Ceftaroline showed efficacy similar to vancomycin in children with cystic fibrosis and pulmonary exacerbations, which are often colonized (and infected) with MRSA [73]. Extensive studies and trials evaluating ceftaroline for treating MRSA/MRSE bacteremia and other invasive infections are lacking. In addition, children and newborns with sepsis often suffer from renal failure and are exposed to other nephrotoxic drugs; therefore, a vancomycin-sparing regimen may ease the management avoiding therapeutic drug monitoring and protecting the renal function. In this view, ceftaroline is a promising agent due to its safety and broad activity spectrum [74]. Further studies may explore its applications in special populations such as oncohematological patients and neonates, as well as for device-related infections (endocarditis and CLABSIs). 

Daptomycin is a novel lipopeptide approved by FDA and EMA for the treatment of cSSSIs and bacteremia caused by Gram-positive bacteria in children older than 1 year, due to its rapid bactericidal effect [75,76]. It is active against MRSA, VRE, and MR-CoNS. Bradley et al. evaluated daptomycin for complicated SSSI in a randomized trial, including 97 MRSA infections treated with daptomycin and 46 with the standard of care, with a similar safety profile. However, children with bacteremia (or other invasive infections), renal insufficiency, or any clinical or laboratory findings suggestive of potential daptomycin toxicity were excluded from this study. The study was not designed to statistically compare the efficacy of daptomycin with the standard of care; however, overall, the efficacy rate was similar between arms [48]. The same authors described results from a randomized trial including eight children with hematogenous MRSA osteomyelitis treated with daptomycin or vancomycin; however, those cases were insufficient to achieve conclusions [21]. Seven children (neonates and infants excluded) with MRSA cSSSIs and one with bacteremia were successfully treated with daptomycin in a prospective Japanese study [39]. On the other hand, we found no trials or prospective studies evaluating the safety and efficacy of daptomycin for VRE infections in children, with publications limited to case series or case reports [22,46,53,60].

Linezolid is the first agent of oxazolidinones and is approved by FDA for adults and children with SSSI or pneumonia, including MRSA or VRE etiologies [77]. In pediatrics, linezolid was studied in a randomized trial, showing clinical efficacy comparable to vancomycin in 20 MRSA severe infections, and showing microbiological eradication for three cases of VRE [78]. In a prospective series of immunocompromised children, linezolid was effective as a single therapy for MRSA and VRE infections without a concomitant worsening of chemotherapy-induced myelotoxicity [23]. Furthermore, linezolid has been described in many pediatric case reports to treat ventriculitis, bloodstream infections, endocarditis, and others [27,49,51,52,56,59,61].

Despite its bacteriostatic activity, linezolid has a favorable efficacy for bacteremia in neonates, comparable to vancomycin [79,80]. Linezolid has the advantages of tolerability for a prolonged course, possible oral shift, and less concern for resistance, particularly for MRSA, compared to vancomycin. 

Two recent meta-analyses showed an increase in the last years of vancomycin-intermediate, resistant, and heterogenous intermediate SA (VISA, VRSA, and hVISA), particularly in the USA and Asia [81,82]. The prevalence of VRSA increased from 1.2% before 2010 to 2.4% after 2010, with the highest frequency in the USA (3.6%) [81]. We found only one case of SA in our research with a MIC of 4 mg/dL for vancomycin [42]. VRE resistance to linezolid is rare but possible. Two case reports described a combination of antibiotics to treat severe VRE infections: linezolid with daptomycin [49], and daptomycin with tigecycline [51]. To date, no recommendations of combination therapy for VRE or other Gram-positive MDR infections have been provided. Concerns exist because the in vitro inhibition of the growth induced by a bacteriostatic drug may reduce efficacy when combined with a bactericidal drug [83]. Clinical data in vivo are missing. 

Tedizolid is a newer oxazolidinone approved by EMA for treating SSSIs in children older than 12 years [84]. In the pediatric population, the safety and efficacy of tedizolid were evaluated in a phase III trial in adolescents with SSSIs caused by multiple pathogens, comparing tedizolid with the standard of care. The study included two MRSA in the tedizolid group and one in the SOC group, but specific outcomes were not reported, and the study was then excluded from our collection [85].

The use of quinupristin/dalfopristin has been described in a retrospective series of children with concomitant comorbidities and serious Gram-positive MDR infection without other drug possibilities [36]. It resulted in clinical success for 71 out of 101 VRE infections (in combination with other antimicrobials). As limitations, quinupristin/dalfopristin is ineffective against *E. faecalis*, requires a central line because it is highly irritant for peripheral veins, and has several adverse effects: metabolic interactions, severe myalgias, arthralgias, nausea, and hyperbilirubinemia [86]. In adults, studies comparing quinupristin to linezolid showed similar efficacy with a better versatility for linezolid [86,87]. In the pediatric population, the paucity of data does not allow us to conclude, but safety concerns led to its progressive desertion in favor of linezolid and daptomycin.

Novel antibiotics are currently approved by the FDA and/or EMA to treat Gram-positive MDROs in adults [88]. Oritavancin is a new-generation lipoglycopeptide, currently approved by the FDA and EMA for acute bacterial soft skin and skin structures infections due to MSSA, MRSA, *Streptococcus* spp., and *E. faecalis* [89]. Despite its potential activity, oritavancin has not been clinically evaluated against VRE. A phase I clinical trial is recruiting pediatric patients with Gram-positive bacterial infections to evaluate the safety and tolerability of oritavancin [90].

Dalbavancin is a semisynthetic lipoglycopeptide with a mechanism of action similar to vancomycin, active against MRSA but not against VRE. It is approved by the EMA and FDA to treat adults with cSSSI [91]. Its long half-life (8.5 days) allows a weekly administration, which would be a massive advantage in pediatrics, as prolonged admissions and vascular catheter management are uncomfortable and expensive. A phase III trial (dalbavancin versus standard of care) recruiting patients aged 3 months to 17 years with SSSI is currently ongoing [92].

### 4.2. MR-CoNS

Compared to other Gram-positive bacteria, there are fewer studies regarding the treatment of MR-CoNS in pediatrics. However, MR-CoNS are widely diffused and poses a challenge, particularly for the limited possibilities in the neonatal population. In our research, neonates were under-represented compared to children. We included only one randomized trial where linezolid was successfully used in 13 neonatal MRSE infections with an 85% eradication rate, compared with vancomycin (100% eradication on six patients). However, no specific information on clinical outcomes was provided in this study [25]. There is a paucity of data regarding daptomycin’s safety and efficacy in neonates, which seems to need higher doses to match the clinically effective exposure [93]. A retrospective case series by Mohzari et al. described 15 preterm infants with MRSE infections treated with daptomycin after vancomycin failure, with a success rate of 11 out of 15 [94].

### 4.3. Neonatal Safety

Safety concerns and the lack of pharmacokinetics data in neonates and preterm neonates explain why antimicrobial research often leaves this population behind in drug approvals. This study was not designed to assess antimicrobial safety in pediatrics; however, due to the paucity of data, we looked at safety outcomes in neonates included in the research. Deville performed a sub-analysis on neonatal patients included in the randomized trial by Kaplan et al., comparing vancomycin and linezolid [79]; the two groups of neonates were homogeneous except for postnatal age (newborns were younger in the linezolid group), and adverse effects were overall similar. Of note, the percentage of patients with abnormal hematology or serum chemistry values was not statistically different in linezolid-treated patients and vancomycin-treated patients. In the case series by Ergaz, three preterm newborns infected with VRE were successfully treated with linezolid without abnormalities in white blood cell count and liver enzymes [62]; the same was observed for the two preterms treated with linezolid presented by Hoehn [52]. A review by Garazzino on the use of linezolid in pediatric and neonates described an excellent safety profile [95]. Daptomycin use was described in the series by Mohzari (median weight 870 g) without reporting muscular or neurologic toxicity in preterms treated for MR-CoNS invasive infections [94]. In the neonatal case reports included in our study, daptomycin was used with good tolerability [64,66].

In the case series by Bradley, 11 neonates treated with ceftaroline for LOS experienced mild adverse effects, except for one serious effect (salmonellosis), but only a case of diarrhea was related to the study treatment [72].

### 4.4. Limitations

This review had some limitations, which were intrinsic to the object of our research. The main limiting factor was the paucity of articles exploring the treatment of Gram-positive in children and neonates; in particular, data about MR-CoNS, albeit widely diffused, are very few. Furthermore, the included studies had different study designs, making it difficult to compare results and excluding the capacity for a meta-analysis. We extrapolated information on MDR pathogens from larger studies that were not designed ad hoc to study them but often described the experience with a single antibiotic. In addition, the quality of evidence was low for most studies due to their retrospective nature (or case reports); the definition of the outcomes (clinical and microbiological success) was not detailed in every study, and it could vary, impairing the strength of the results.

## 5. Conclusions

In conclusion, robust evidence on the treatment of MRSA, VRE, and MR CoNS is lacking in the pediatric and neonatal populations. However, a trend toward newer and safer molecules is observed (ceftaroline and linezolid) compared to the standard of care (vancomycin) for MRSA. Further studies are needed to investigate their effectiveness in particular settings (immunocompromised children, device-related infections, and critical care settings). Linezolid seems promising in the neonatal population for its activity against MRSA and MR CoNS, safety profile, and optimal bioavailability. Daptomycin and linezolid remain the cornerstones of VRE treatment. Novel molecules (oritavancin and dalbavancin) are currently being studied in pediatrics, with promising applications.

## Figures and Tables

**Figure 1 antibiotics-12-00261-f001:**
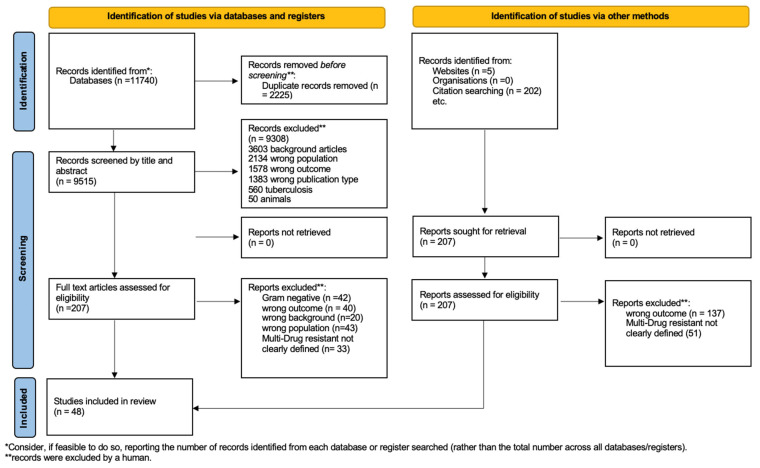
Flowchart of the study selection process.

**Figure 2 antibiotics-12-00261-f002:**
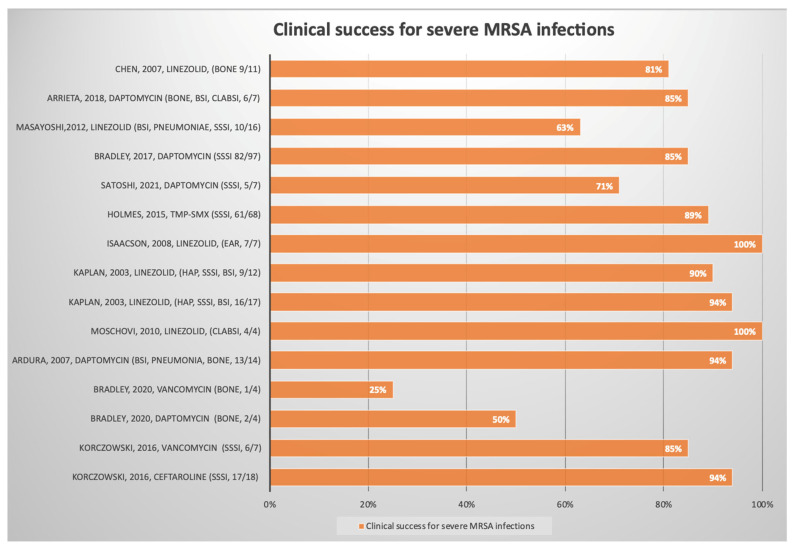
Results regarding the choice of treatment and clinical success for methicillin-resistant *Staphylococcus aureus* (MRSA) infections. Clinical success was defined as complete resolution or substantial improvement of the signs and symptoms of the index infection. BSI: bloodstream infection; SSSI: skin and structure skin infections, CLABSI: central line-associated bloodstream infection, HAP: hospital-acquired pneumonia.

## Data Availability

The data presented in this study are available in article and in Appendix A.

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
