# Peer review of "Therapeutic Options and Outcomes for the Treatment of Children with Gram-Positive Bacteria with Resistances of Concern: A Systematic Review"

_antibiotics, 2023, doi:10.3390/antibiotics12020261_

Round 1

Reviewer 1 Report

In this systematic review, the authors have followed PRISMA guidelines and searched articles found in the database for treatment performed by physicians in children with MRSA, VRE, and MR-CoNS infection. They found an increasing trend of using linezolid, daptomycin, and ceftaroline for pediatric and neonatal populations. Recently, the authors published an article in Antibiotics Journal regarding therapeutic options and outcomes for treating neonates and preterms with gram-negative MDR Bacteria (10.3390/antibiotics11081088), and this article follows the same search strategy with treatment against MDR gram-positive bacteria in children. The current manuscript contains about 14% similarity with the published one, which needs to be carefully modified. Although the authors found some common groups of antimicrobials suggested by the physicians for the MDR gram-positive cases, data was extracted from only 11 articles that were assessed as good quality by the authors, and about 26 articles discussed only one patient. It would be better if the article could be merged with their previous article on gram-negative bacteria. The lack of articles fulfilling the study's objective makes it challenging to draw any conclusions. However, the authors can modify the search strategy by adding fungal and viral infections along with gram-positive bacteria and enriching the manuscript. 

Author Response

In this systematic review, the authors have followed PRISMA guidelines and searched articles found in the database for treatment performed by physicians in children with MRSA, VRE, and MR-CoNS infection. They found an increasing trend of using linezolid, daptomycin, and ceftaroline for pediatric and neonatal populations. Recently, the authors published an article in Antibiotics Journal regarding therapeutic options and outcomes for treating neonates and preterms with gram-negative MDR Bacteria (10.3390/antibiotics11081088), and this article follows the same search strategy with treatment against MDR gram-positive bacteria in children. The current manuscript contains about 14% similarity with the published one, which needs to be carefully modified.

Thank you for reply, we used the same search strategy, but we included articles in which were possible to extrapolate data for Gram positive bacteria. I think that 14% of similarity could be referable only for the Methods. However, we tried to modify the paper as much as possible.

Although the authors found some common groups of antimicrobials suggested by the physicians for the MDR gram-positive cases, data was extracted from only 11 articles that were assessed as good quality by the authors, and about 26 articles discussed only one patient. It would be better if the article could be merged with their previous article on gram-negative bacteria. The lack of articles fulfilling the study's objective makes it challenging to draw any conclusions. However, the authors can modify the search strategy by adding fungal and viral infections along with gram-positive bacteria and enriching the manuscript.

Thank you for your comment. This is a very important advice for us, but we had divided our research in two articles to explain better the different between two infectious disease groups. The lack of hard studies and the paucity of data are an itself evidence. Gram positive, negative, fungi and virus have different epidemiology and treatment and it this not simple to condense the arguments in a single article.

Reviewer 2 Report

I would like to thank the authors for their efforts in conducting this review. Few comments to consider. 

Title:

I suggest to edit the wording of the title to be: Therapeutic Options and Outcomes for the Treatment of Children with resistant Gram-Positive Bacteria: A Systematic Review

Abstract:

MDR, use the full word first time used

If you use genus and species name (Staph aureus) make italic

Introduction:

Please limit the abbreviations specially for those that will not be used again in the manuscript

What is the definition of MDR organisms?

Methods:

How about searching in Scopus?

You are searching only for Gram-positive, why did noy you use Gram-positive as one of the mesh terms?

Instead of “MDR” as a mesh term, you should have used “resistant”

Did you register the study in PROSPERO?

Figure 1

MDR use full word or define underneath the figure

Results:

High income countries: use the abbreviation after first encounter

Figure 2:

Define the abbreviations underneath the figure (CLABSI, HAP, etc.)

What is the definition of clinical success? Are all these studies used the same definition?

Author Response

I would like to thank the authors for their efforts in conducting this review. Few comments to consider.

Thank you for the suggest

Title:

I suggest to edit the wording of the title to be: Therapeutic Options and Outcomes for the Treatment of Children with resistant Gram-Positive Bacteria: A Systematic Review

Thank you for the comment. We think that is important underling the concept “with resistances of concern”, not only resistant. For this, we prefer to maintain this title. We underlined this concept because we had analyzed more commonly bacteria that caused infection in clinical practice, and not overall resistant Gram positive bacteria.

Abstract:

MDR, use the full word first time used

Thank you for the suggest

If you use genus and species name (Staph aureus) make italic

Thank you we will correct

Introduction:

Please limit the abbreviations specially for those that will not be used again in the manuscript

We thank you for the comment

What is the definition of MDR organisms?

Thank you for the suggest. We followed international expert proposal (Multidrug-resistant, extensively drug-resistant and pandrug-resistant bacteria: an international expert proposal for interim standard definitions for acquired resistance, Magiorakos et alt. Clin Microbiol Infect 2012). This proposal considered the resistant related on bacterial species (Staphilococcus aures, entococcus spp. etc.) and the definition of Multi Drug resistant strictly understood could not be applied to analyzed bacteria: CoNS are not included in MDR definition; Enterococcus spp. are considered if they are resistant to 2 or more classes and MRSA are considered MDR as such. Therefore, we did not define MDR bacteria for this reason.  

Methods:

How about searching in Scopus?

Thank you, we will keep in mind this suggestion, however for this research we used OVID search system that include database for MEDLINE and Embase. We searched on Cochrane library separately.

You are searching only for Gram-positive, why did noy you use Gram-positive as one of the mesh terms?

Thank you, we will add the mesh terms.  

Instead of “MDR” as a mesh term, you should have used “resistant”

Thank you for the suggest, we will modify.

Did you register the study in PROSPERO?

Yes, we will include in METHODS the registration statement

Figure 1

MDR use full word or define underneath the figure

Thank you, we will modify the figure 1

Results:

High income countries: use the abbreviation after first encounter

Thank you for the suggest.

Figure 2:

Define the abbreviations underneath the figure (CLABSI, HAP, etc.)

Thank you for the suggest, we add the definition

What is the definition of clinical success? Are all these studies used the same definition?

Thank you, we defined clinical success in METHODS section. In major of studies, definition of clinical success is clarified, and it is considered how clinical healing

Reviewer 3 Report

Dear Editor,

Many thanks for inviting me to review this paper. This study Therapeutic Options and Outcomes for the Treatment of Children with Gram-Positive Bacteria with resistances of concerns. I write my suggestions below.

I believe this study aligns with the scope of the journal. Antibiotics is a highly reputable academic journal and have a distinguished audience. And its’ audience deserves high-quality and exquisite publications.

To increase the impact of the paper and readability authors should work on how to present the novelty and impact of the paper. There are some limitations of the study. I believe this study should be reassessed after the minor issues are resolved.

·                I believe the title is suitable and adheres to the content of the study.

·                PICO info should be given in the abstract.

·                How did the last sentence was added to the abstract section? What is the relationship of the abstract overall?

·                The take-home message is quite broad in the abstract section. I believe detailed outcomes should be underlined.

·                Keywords: I would like to recommend the adhere MeSH headings.

·                I believe the introduction section is well organized and beneficial.

·                Please adhere to the journal guideline, especially for reference sections. Need to adhere to the guidelines for authors.

·                It would be better to give up-to-date references.

·                The time span of the included is rather broad. What was the rationale for including the last 22 years? Usually in SR time span of the last 5-10 years is included.

·                Usually in SR an even number of researchers decide the inclusion or exclusion of the papers. Hence in your study, the final decision has been made by 4 reviewers how did the disagreement handle when 2 reviewers opposed and 2 accepted?

·                The study types of the included studies are extremely broad. How did the authors handle the biases emerged due to study type? Since this is a SR, a comparison usually expected at the end of study since this study included studies with different methodological designs an explanation about handling the bias is necessary.

·                Quality assessment method is rather complicated to understand. Could the authors clarify the method and the presentation of the quality assessment of the included studies?

·                In figure 1 Which databases how many papers authors gathered? Giving the detailed numbers would be helpful. How come none of the studies were excluded due to bias?

·                I would like to suggest authors give some details about the outcomes.

·                Discussion is quite long and hard to follow. Please sum it up accordingly.

· Limitations should be more explored.

Author Response

Dear Editor,

Many thanks for inviting me to review this paper. This study Therapeutic Options and Outcomes for the Treatment of Children with Gram-Positive Bacteria with resistances of concerns. I write my suggestions below.

I believe this study aligns with the scope of the journal. Antibiotics is a highly reputable academic journal and have a distinguished audience. And its’ audience deserves high-quality and exquisite publications.

To increase the impact of the paper and readability authors should work on how to present the novelty and impact of the paper. There are some limitations of the study. I believe this study should be reassessed after the minor issues are resolved.

  • I believe the title is suitable and adheres to the content of the study.

Thank you for the comments.

  • PICO info should be given in the abstract.

Thank you for the advice. For brevity reasons we did not include PICO in the abstracts, but we will include it in the METHODS.

  • How did the last sentence was added to the abstract section? What is the relationship of the abstract overall?

Thank you for suggestion. We found new molecules under investigation by searching articles about Gram positive Bacteria. However, these articles were out of the scope of our research, as they did not fulfil inclusion criteria so we will remove this sentence from the abstract.

  • The take-home message is quite broad in the abstract section. I believe detailed outcomes should be underlined.

Thank you, we will modify the sentence

  • Keywords: I would like to recommend the adhere MeSH headings.

Thank you, we will check to adhere Mesh heading

  • I believe the introduction section is well organized and beneficial.

Thank you for the comment

  • Please adhere to the journal guideline, especially for reference sections. Need to adhere to the guidelines for authors.

Thank you for reply. We took care references and authors according journal guideline.

  • It would be better to give up-to-date references.

Thank you for your suggestion.

  • The time span of the included is rather broad. What was the rationale for including the last 22 years? Usually in SR time span of the last 5-10 years is included.

Thank you for the comment. Pediatric studies progress slower than adult studies. We amplified the period of research to include much more studies.

  • Usually in SR an even number of researchers decide the inclusion or exclusion of the papers. Hence in your study, the final decision has been made by 4 reviewers how did the disagreement handle when 2 reviewers opposed and 2 accepted?

Thank you for the comment. We declared the reference of our strategy assessments: “Assessments of the titles, abstracts, and full texts were conducted independently by three investigators (L.C., C.L. and L.R.). Discussion with a fourth reviewer (D.D.) resolved any disagreement regarding study selection.” The fourth is senior reviewer and he resolved the disagreement, if presenting, with the other three reviewers. We will specify this in METHODS.

  • The study types of the included studies are extremely broad. How did the authors handle the biases emerged due to study type? Since this is a SR, a comparison usually expected at the end of study since this study included studies with different methodological designs an explanation about handling the bias is necessary.

Thank you for the suggest. Our research respected the systematic revision criteria, but results exclude the possibility of a comparison and meta-analysis. This limitation is often presented in pediatric studies.

  • Quality assessment method is rather complicated to understand. Could the authors clarify the method and the presentation of the quality assessment of the included studies?

Quality assessment was conducted according to Study Quality Assessment Tool from The National Heart, Lung, and Blood Institute We will define better in the paper. We will clarify better the assessment method in METHODS.

  • In figure 1 Which databases how many papers authors gathered? Giving the detailed numbers would be helpful. How come none of the studies were excluded due to bias?

Thank you, unfortunately we cannot deduce how much studies are been extracted from single database.

  • I would like to suggest authors give some details about the outcomes.

Thank you, we will add more details about outcome in METHODS

  • Discussion is quite long and hard to follow. Please sum it up accordingly.

Thank you for the suggest, we have reduced the text. For comprehensive clarity, if the editor allows it, we have divided it into subsections, to keep all the information about old and new molecules that we have obtained from the research.

  • Limitations should be more explored.

Thank you for the suggest, we will explore better

Round 2

Reviewer 1 Report

The authors have tried to address the comments, but the lack of data to answer their research question wasn't enough. However, the authors tried to address this issue in the limitation section.